# Mendelian Randomization Indicates a Causal Role for Omega-3 Fatty Acids in Inflammatory Bowel Disease

**DOI:** 10.3390/ijms232214380

**Published:** 2022-11-19

**Authors:** Courtney Astore, Sini Nagpal, Greg Gibson

**Affiliations:** Center for Integrative Genomics, School of Biological Sciences, Georgia Institute of Technology, Krone EBB1 Building, 950 Atlantic Drive, Atlanta, GA 30332, USA

**Keywords:** polygenic risk, Crohn’s disease, docasahexaenoic acid, UK BioBank

## Abstract

Inflammatory bowel disease (IBD) is characterized by chronic inflammation of the gastrointestinal system. Omega-3 (ω_3_) fatty acids are polyunsaturated fatty acids (PUFAs) that are largely obtained from diet and have been speculated to decrease the inflammatory response that is involved in IBD; however, the causality of this association has not been established. A two-sample Mendelian randomization (MR) was used to assess genetic associations between 249 circulating metabolites measured in the UK Biobank as exposures and IBD as the outcome. The genome-wide association study summary level data for metabolite measurements and IBD were derived from large European ancestry cohorts. We observed ω_3_ fatty acids as a significant protective association with IBD, with multiple modes of MR evidence replicated in three IBD summary genetic datasets. The instrumental variables that were involved in the causal association of ω_3_ fatty acids with IBD highlighted an intronic SNP, rs174564, in *FADS2*, a protein engaged in the first step of alpha-linolenic acid desaturation leading to anti-inflammatory EPA and thence DHA production. A low ratio of ω_3_ to ω_6_ fatty acids was observed to be a causal risk factor, particularly for Crohn’s disease. ω_3_ fatty acid supplementation may provide anti-inflammatory responses that are required to attenuate inflammation that is involved in IBD.

## 1. Introduction

The human metabolome gauges environmental factors and genetic influences that can be indicative of abnormal biochemical and/or physiological health [1]. While IBD has been previously thought to only impact the Western world, the occurrence is emerging in developing countries, possibly due to modern environmental exposures [2]. Approximately 1.4 million Americans are impacted by IBD, which may be the result of environmental exposures such as diet and smoking status [3]. Identification of environment-metabolite-disease associations may aid in the discovery of diagnostic biomarkers or therapeutic targets, including simple dietary interventions for conditions such as inflammatory bowel disease (IBD), which is a complex gastrointestinal disease that is characterized by immune dysregulation, microbiota alterations, and intestinal permeability. Metabolomic profiles have been previously used to discriminate IBD patients from controls. It has also been shown that the gut microbiome can impact metabolite levels, especially those of individuals with IBD [3,4]. IBD has a complex pathogenesis with multiple genetically and phenotypically distinct subtypes, including Crohn’s disease (CD) and ulcerative colitis (UC). The complexity of IBD pathogenesis also contributes to the multiplicity of possible treatment options [5]. There are currently four classes of approved drugs with hundreds of molecules undergoing various phases of the drug discovery process for IBD [6]. It has also been realized that there are modifiable environmental factors such as diet that can benefit the treatment and management of IBD [7,8].

Omega-3 (ω_3_) fatty acids are a subtype of polyunsaturated fatty acids (PUFA) which are fat molecules having more than one unsaturated carbon bond in the backbone of the structure. There are three notable forms of ω_3_ fatty acids, alpha-linolenic acid (ALA), eicosapentaenoic acid (EPA), and docasahexaenoic acid (DHA). Each type of ω_3_ fatty acid differs slightly in its chemical structure and source; ALA is obtained mostly from plant oils while EPA and DHA are obtained primarily from fish/seafood. ALA is not synthesized in the human body and can only be obtained through diet. However, the body can convert small amounts of ALA into EPA and then to DHA, in a series of steps catalyzed by the *FADS2* and *FADS1* desaturase enzymes [9]. An extensive literature has investigated the association of ω_3_ fatty acids with health benefits [10,11] including a likely reduction of the inflammatory responses that are seen in individuals with IBD, although the genetics and causal relationship remain unclear [12,13].

To assess the possible causal associations between ω_3_ fatty acids and IBD, we herein employ Mendelian randomization (MR) [14]. This approach takes advantage of the invariant nature of genotypes in individuals which can be used as instruments to assess whether an exposure (such as a metabolite) mediates the condition. Essentially, we regress genotypic disease effects on all highly significant metabolite effects that were obtained from genome-wide association studies. MR is advantageous as there are multiple methods for adjusting for confounding influences, and hence assessing causal associations, and one can also assess the direction of effect of the relationship (protective or risk) [15]. A recent MR study assessed the causal association between ω_6_ fatty acids and IBD [16], concluding that anti-inflammatory PUFAs downstream of *FADS1* may be protective. Both pro- and anti-inflammatory mechanisms have been linked with increased levels of various PUFAs; however, such mechanisms remain challenging to unveil [17,18]. Our study focuses on the causal relationship between ω_3_ fatty acids and IBD, with implications regarding their effect on other autoimmune diseases such as rheumatoid arthritis [19,20].

## 2. Results

### 2.1. Causal Association between Circulating ω_3_ Fatty Acids and IBD

To evaluate whether ω_3_ fatty acids are likely to be causally involved in the pathogenesis of IBD, we identified 31 IBD-associated SNPs in the Sample one IBD-Genetics Consortium (GC) GWAS (Appendix A), 38 SNPs in the Sample two IBD-GC GWAS (Appendix A), and subsequently 43 SNPs in the Sample three IBD-FinnGen GWAS (Appendix A) as genetic instruments. Using the 31 sample one SNPs, which had an average F-statistic of 10, implying strong confidence as instrumental variables overall, it was observed that increased circulating ω_3_ fatty acids are likely to mediate decreased risk for IBD (IVW OR 0.87; 95% CI: 0.82–0.92; *p* = 6.89 × 10^–6^). The *p*-values for the other MR methods (i.e., MR-Egger, weighted median, simple mode, and weighted mode), were all significant with *p* < 10^−3^ and with ORs less than 1.0, also consistent with a protective role of ω_3_ fatty acids. Replicating this result for the earlier release of the sample two IBD-GC GWAS weights as genetic instruments, which had an average F-statistic of 7.4, it was further observed that ω_3_ fatty acids resulted in a decreased risk for IBD (IVW OR 0.79; 95% CI: 0.75–0.89; *p* = 7.42 × 10^−6^). The *p*-values for the other MR methods (i.e., MR-Egger, weighted median, simple mode, and weighted mode), were consistently less than 0.01, and the ORs were less than 1.0. In addition, the independent replication Sample 3 IBD GWAS from FinnGen, for which the SNPs had an average F-statistic of 6.0, it was also observed that ω_3_ fatty acids resulted in a decreased risk for IBD (IVW OR 0.86; 95% CI: 0.77–0.96; *p* = 6.9 × 10^−3^). The ORs were less than one across all methods with *p*-values < 0.01, except for the simple mode MR method. The slopes representing the causal association of each MR method are shown in Figure 1 for ω_3_ fatty acids in the Sample one and Sample two IBD-GC GWAS datasets.

When removing the SNPs with an F-statistic < 10 in Sample one, the *p*-values remained in the range of 10^−5^ < *p* < 10^−2^ with ORs less than 1.0. By Fisher’s method for combining *p*-values from all three samples, all MR modes were significant at *p* < 0.0001, except for the simple mode (*p* < 0.01). Full summary statistics of each MR method are shown in Table 1.

Given the heterogeneity of ω_3_ fatty acid effect sizes, with one third of the SNPs showing nominal effects in one of the samples in the opposite direction to the overall trend (that is, with positive effects on disease) we performed multiple sensitivity analyses. However, neither Cochran’s Q test *p*-value for the MR-Egger and IVW methods nor the MR-Egger intercept *p*-value resulted in *p* > 0.05, indicating no strong evidence for horizontal pleiotropy involving the genotypes that were used for either IBD GWAS dataset (Table 1). Similarly, leave-one-out analysis indicated the same direction of effect of the causal association despite removal of each genetic instrument for all three IBD GWAS datasets (Appendix A showing the result for Sample one IBD GWAS). These results imply that the inferred protective effect of ω_3_ fatty acids for IBD is attributable to multiple variants, not just the major-effect SNP in *FADS2*.

### 2.2. Regulatory Effects of the Genetic Instruments in Blood and Colon

To identify which tissue the effect may be mediated in, we examined the eQTL profiles for all SNPs used as ω_3_ fatty acid genetic IVs for the IBD GWAS. The regulatory effects on gene expression, namely eQTLs, were extracted from the GTEx database for sigmoid colon, transverse colon, and whole blood. There were some instances where a SNP had multiple corresponding genes, but just nine of the SNPs that did not have an eQTL in any of the specified tissues. Comparing the effect sizes of the eQTLs between the sigmoid and transverse colon to whole blood, we observed consistent positive correlation (Figure 2), implying that the genotypes are likely to have similar influences on ω_3_ fatty acid abundance in all three tissues.

Notably, the largest eQTL effect sizes were consistently observed for *FADS2*, which encodes a fatty acid desaturase, and showed the largest GWAS association with IBD. Since *FADS1* has also been implicated in IBD risk [16], we further contrasted the eQTL effects of the lead GWAS variant at the *FADS1*/2 locus, rs174564, on both genes, finding a much stronger association with *FADS2* in blood and terminal ileum, and similar effect sizes but in opposite directions in the colon (Appendix A). Intriguingly, the sign of the regulatory effect is also reversed in the liver, where *FADS1* is more strongly influenced. Other consistently large associations were observed for *DOCK6* and *DOCK7* (which are located on different chromosomes), *CATSPER2*, *INCENP*, and *ZMIZ2*, the latter in the opposite direction with respect to the effect of the alternate allele on expression. *LPL* (encoding lipoprotein lipase) appears to be a blood-specific factor, whereas *MAU2*, *DAGLA* (encoding diacylglycerol lipase alpha), and *SIK3* may have opposite signs in the sigmoid and transverse colon. These results suggest that further investigation of the role of some of these genes in fatty acid metabolism may illuminate the mechanisms of association with IBD. 

### 2.3. Causal Association of Additional Metabolites for IBD

MR screening of 249 metabolite measures as exposures revealed a total of five additional metabolite measures that have evidence for a causal association with IBD according to the primary MR method, IVW. These were concordant at the multiple testing threshold of 10^−4^ in both the Sample one IBD GWAS and Sample two IBD GWAS datasets as shown in the volcano plots in Figure 3A,B, respectively.

The same measures were also the most significant in the FinnGen replication Sample three, although only at the nominal *p* < 0.05 level. Of the five metabolite measures, four were protective (OR < 1.0) and one was a risk factor (OR > 1.0). Notably, the concordant protective associations all involve ω_3_ fatty acids in some capacity: Docosahexaenoic acid (DHA, the most unsaturated of the lipids), the ratio of DHA to total fatty acids, and the ratio of ω_3_ fatty acids to total fatty acids. The risk factor that is concordant with both the IBD discovery and replication was the ratio of ω_6_ fatty acids to ω_3_ fatty acids, corroborating the protective role of ω_3_ fatty acids. One source of DHA is ghee, which has been previously speculated to aid in the prevention of some diseases [21]. 

We also considered whether ω_3_ fatty acids interact with genetic risk to influence IBD prevalence in the UK Biobank. While ω_3_ fatty acid, and particularly DHA, levels are significantly reduced in IBD cases relative to the controls, the difference is similar across the spectrum of the IBD polygenic risk score (Figure 4). Similarly, we did not observe any statistical interaction between rs174564 and DHA or ω_3_ fatty acids and IBD prevalence, implying that genetic and dietary influences have additive effects on risk. It is noteworthy that smoking status, type of bread consumption, and alcohol intake all do strongly interact with polygenic risk, as does fresh fruit intake, but fish consumption does not [22].

## 3. Discussion

The positive causal association between ω_3_ fatty acids and IBD was investigated. From this, ω_3_ fatty acids were identified as one of the most significant protective associations with IBD, with MR evidence in the range of 10^−6^ < *p* < 10^−2^ across all five MR methods in three IBD GWAS samples. This study enhances the findings in [16] regarding PUFA involvement by establishing that it is ω_3_ rather than ω_6_ fatty acids that are likely to be protective, and implicating *FADS2* over *FADS1* as the key enzyme in the small bowel specifically influencing Crohn’s disease risk. We did not find any evidence of a significant bidirectional effect of IBD on ω_3_ fatty acids.

Multiple lines of evidence support the robustness of our findings. First, the MR results are replicated in two versions of the IBD-GC’s large-scale GWAS that produced different SNP effect weights, and in the independent FinnGen IBD study. Second, five different implementations of two-sample MR that each filter different SNPs based on certain criteria, all support the inference that the ω_3_ fatty acid-associated genetic instruments jointly exert protective effects. Third, the leave-one-out approach shows that the result is not driven solely by the very strong effect of rs174564, since the exclusion of that variant reduces the significance of the result but does not eliminate it. Also, the core result remains significant when just the five SNPs with F-statistics greater than 10 are used. Fourth, individuals with IBD have lower levels of ω_3_ fatty acid levels and DHA levels than that of controls in the UK BioBank. 

The mechanism by which ω_3_ fatty acids protect against IBD remains to be clarified. They are generally thought to be anti-inflammatory, but some studies also indicate a pro-inflammatory function, for example in rheumatoid arthritis joints [19]. It is thus important to experimentally evaluate the effect of PUFAs on diverse immune cell subsets in the context of tissue residency. Previous studies have shown that ω_3_ fatty acids inhibit cyclooxygenase (COX), which produce prostaglandin hormones that have pro-inflammatory effects [23]. This also indicates that ω_3_ fatty acid-mediated COX inhibition resembles the mechanism of action of aspirin [23]. Modification of hormone and cytokine production is also thought to polarize lymphoid and myeloid subsets, which could have a variety of local and systemic effects. The major genetic risk factor in African Americans [24] 2 *PTGER4*, the receptor for prostaglandin e2, raising the possibility that dietary effects may be strongly mediated through PUFAs to influence disease in this population.

This study implicates several IBD-associated loci as being involved in fatty acid metabolism, but the molecular mechanisms remain to be elucidated. Of particular interest are the pair of guanine nucleotide exchange factor-encoding genes, *DOCK6* and *DOCK7*, which are better known for their roles in the regulation of cytokinesis in neurite outgrowth, skin, and bone development. The genes are unlinked and hence are clearly independent associations operating in similar directions with respect to ω_3_ fatty acid abundance and IBD risk. *SIK3* encodes a kinase that is implicated in mTOR signaling, and *ZMIZ2* is a transcription factor that enhances androgen receptor signaling, but *MAU2* is engaged in sister chromatid cohesion; these may not be the causal genes at the loci. Associations with multiple lipoproteins also deserve further investigation.

One gene whose role in ω_3_ fatty acid production is more direct is the one that is adjacent to the strongest genetic instrument, the intronic SNP, rs174564, in *FADS2*, a desaturase protein engaged in the first desaturation step of alpha-linolenic acid metabolism from C18:3 to C18:4. Decreased *FADS2* expression has been previously linked to the pro-inflammatory response in IBD [11]. A recent MR study [18] showed that this SNP associates with risk of rheumatoid arthritis and is mostly responsible for the inferred causal role of ω_3_ fatty acids in promoting that autoimmune condition since it drove those MR results. We also observed that FADS2 expression is elevated in both the blood and colon in individuals with the IBD-protective variant, consistent with a key role for that enzyme in both tissue compartments. Notably, rs174564 was identified as an eQTL for *FADS1* as well, although with different directions of effect in comparison to *FADS2* for some tissues. In addition, we found that the effect allele for rs174564 was attributed with decreased levels of ω_3_ fatty acid and DHA levels in the UKBB as shown in Appendix A. Further studies are needed to elucidate the reasons for the inverted effects on IBD and RA, and to establish whether these also explain some of the reported negative health effects of taking an excess of ω_3_ fatty acid supplements.

Despite the compelling evidence for a causal role of ω_3_ fatty acid in protection against IBD, our study has several limitations. First, although each of the genetic instruments is genome-wide significant, and hence robustly associated with ω_3_ fatty acid levels in the UK Biobank study, it would be desirable to confirm these effects in more diverse cohorts, particularly with extensive African, Asian, or Hispanic ancestry. Second, although we see a strong correlation in eQTL effects between blood and colon, the study does not establish which tissue type is the more important mediator. Since the protection against ulcerative colitis appears to be much weaker than for Crohn’s disease, a specific colonic role is implicated, but note that the metabolites were measured circulating in serum. The MR results for ω_3_ fatty acids and Crohn’s disease were significant prior to genetic instrument outlier detection via IVW-radial, but not formally significant with outlier detection, and also not significant for ulcerative colitis as shown in Appendix A. Third, although two-sample Mendelian randomization is a powerful discovery tool, one sample MR would confirm the findings and allow for direct evaluation of confounders and covariates that may contribute to pleiotropic influences. The inclusion of ω_3_ fatty acid measurement in ongoing IBD GWAS would facilitate such studies.

However, these limitations should not detract from the conclusion that dietary supplementation of ω_3_ fatty acids ought to be reconsidered as a therapeutic option. Our results only directly inform the prevalence of the disease and should not be taken to imply that ω_3_ fatty acid supplementation will be curative. It may, for example, need to be administered for an extended period, or at a particular age, and likely only benefits a subset of patients. Clear differences in *FADS2* haplotypes across ancestries have been proposed to explain variable results in clinical trials involving ω_3_ fatty acid supplements [25], and dietary shifts may feasibly contribute to population differences in the rates of increase in IBD prevalence [26].

## 4. Materials and Methods

### 4.1. Study Design

A two-sample MR was employed to assess the causal associations between genetically predicted ω_3_ fatty acid levels as exposures and IBD as the outcome [27]. MR was performed to evaluate the hypothesis that the selected genetic instruments, which predict the modifiable exposure ω_3_ fatty acids, are causally associated with the exposure-related outcome [28]. Stringent criteria for genetic instrument selection, as described in the Appendix A, were implemented in the open source 2SampleMR package in R [29], also supplemented with additional controls. The study design is shown in Figure 5 and includes two releases of the IBD-GC GWAS summary statistics [30,31] and an independent FinnGen GWAS [32] for IBD associations, as well as metabolome GWAS summary statistics from the UK Biobank.

We followed the STROBE-MR guidelines for performance and reporting of MR studies [33,34]. Details on the summary statistics are provided in the Appendix A online, along with the criteria that was used for genetic instrument selection.

### 4.2. Statistical Analysis

There were three methods that were performed to assess the potential impact of pleiotropy involving the selected ω_3_ fatty acid genetic instruments [35]: Cochran’s Q test [36], MR-Egger intercept test [37], and leave-one-out analysis [38], also explained in more detail in the Supplement [39]. A total of five MR methods were used to assess the causal relationship between metabolites and IBD. The IVW method was utilized as the primary method as it assumes the validity of all genetic instruments, while the other four (MR-Egger, weighted median, simple mode, and weighted mode) serve as sensitivity checks [30]. The IVW-radial method was used for outlier detection and removal [40]. These ensure that the causal association is replicated across multiple MR methods despite their differences in SNP selection for generating the regression [41].

### 4.3. Gut vs. Blood eQTL Analysis

To further explore the possible functional impact of the genetic instruments determined by MR to mediate the causal relationship between ω_3_ fatty acids and IBD, whole blood, transverse colon, and sigmoid colon eQTL normalized effect sizes and *p*-values were extracted from the Genotype-Tissue Expression (GTEx) database for each of the gene-SNP pairs using the GTEx eQTL calculator [42]. The nearby genes for each SNP were annotated via ANNOVAR [43] Notably, some of the SNPs had an association with the expression of multiple genes, in which case every possible pair was included to minimize bias. Whole blood, gut, and liver eQTL normalized effect sizes were compared to assess the tissue specificity of the lead *FADS1*/2 SNP.

## Figures and Tables

**Figure 1 ijms-23-14380-f001:**
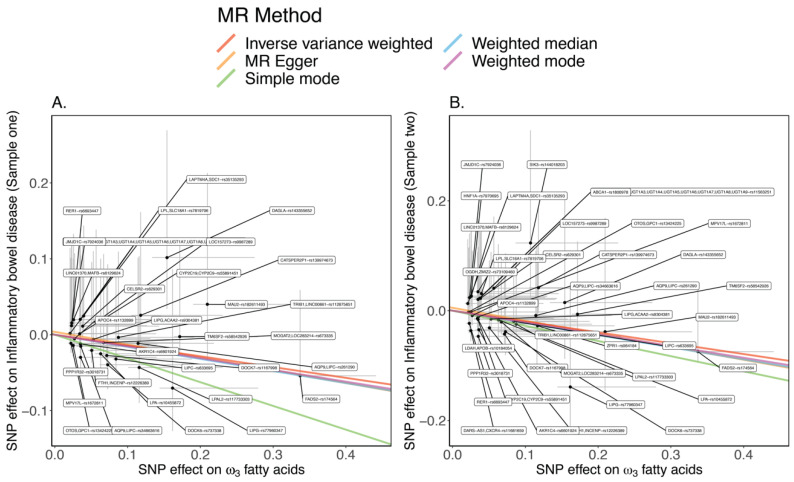
(**A**) Slopes representing the causal association for each MR method for ω_3_ fatty acids on the Sample one IBD GWAS. (**B**) Slopes representing the causal association for each MR method for ω_3_ fatty acids on the Sample two IBD GWAS. Each point is labeled with the SNP and nearby gene(s).

**Figure 2 ijms-23-14380-f002:**
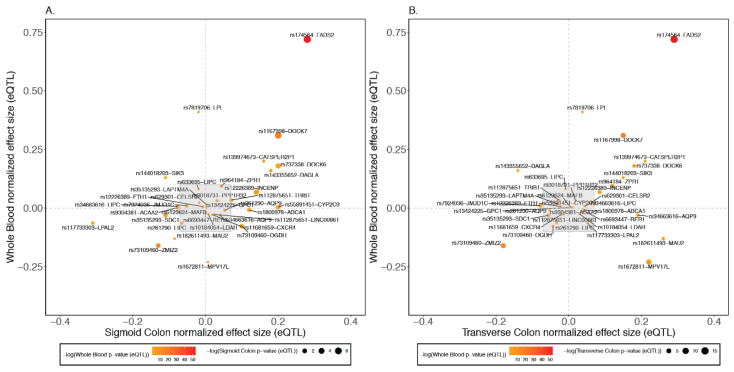
(**A**) eQTL effect size comparisons between the sigmoid colon vs. whole blood. The color of the points represent the −log10(whole blood *p*-value (eQTL)) and size of the points represent −log10(sigmoid colon *p*-value (eQTL)). (**B**) eQTL effect size comparisons between the transverse colon vs. whole blood. The color of the points represent the −log10(whole blood *p*-value (eQTL)) and size of the points represent −log10(transverse colon *p*-value (eQTL)). The grey boxes indicate SNPs that generally have eQTL effects with *p* > 0.05 in both tissues.

**Figure 3 ijms-23-14380-f003:**
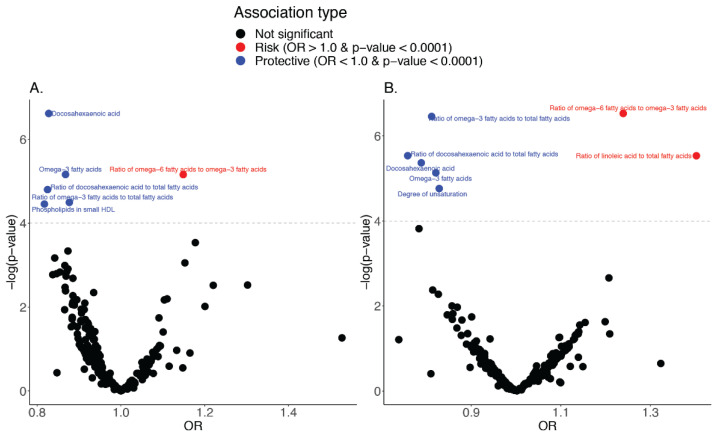
(**A**) IVW method MR results of screening of 249 metabolites as exposures on IBD (Sample one). (**B**) IVW method MR results of screening of 249 metabolites as exposures on IBD (Sample two). Each point represents the causal relationship between a metabolite and the outcome. Red points indicate that the metabolite is a risk factor for IBD, and blue points indicate that the metabolite is protective for IBD. The dashed grey line represents the multiple testing *p*-value threshold of 1.0 × 10^−4^ (0.05/2 × 249).

**Figure 4 ijms-23-14380-f004:**
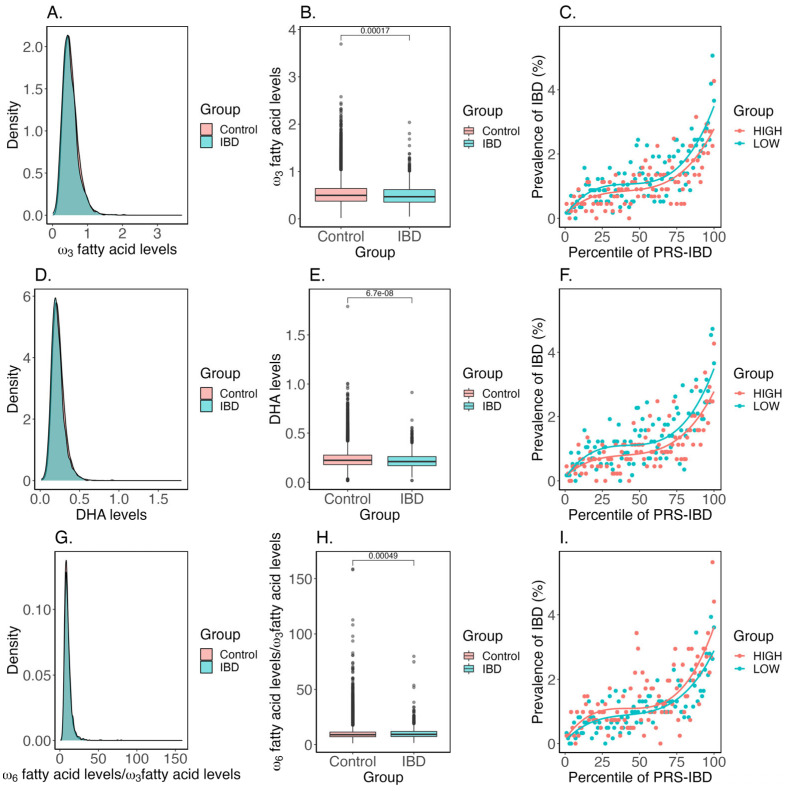
(**A**) Density plot comparing IBD case vs. control ω_3_ fatty acid levels. (**B**) Box plot comparing IBD case vs. control ω_3_ fatty acid levels. The number above the boxes represents the Welch’s *t*-test *p*-value. (**C**) The prevalence vs. risk of IBD for high vs. low ω_3_ fatty acid levels. (**D**) Density plot comparing IBD case vs. control DHA levels. (**E**) Box plot comparing IBD case vs. control DHA levels. The number above the boxes represents the Welch’s *t*-test *p*-value. (**F**) The prevalence vs. risk of IBD for high vs. low DHA levels. (**G**) Density plot comparing IBD case vs. control ω_6_ fatty acid to ω_3_ fatty acid ratio. (**H**) Box plot comparing IBD case vs. control ω_6_ fatty acid to ω_3_ fatty acid ratio. The number above the boxes represents the Welch’s *t*-test *p*-value. (**I**) The prevalence vs. risk of IBD for high vs. low ω_6_ fatty acid to ω_3_ fatty acid ratio.

**Figure 5 ijms-23-14380-f005:**
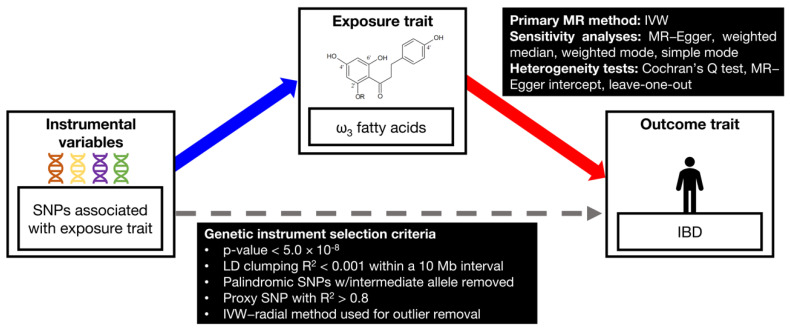
Methodology depicting the MR study design for selecting ω_3_ fatty acid genetic instruments and assessing the causal relationship between circulating ω_3_ fatty acids on IBD.

**Table 1 ijms-23-14380-t001:** MR results for ω_3_ fatty acid IVs on IBD (Samples one, two, and three).

MR Method	IBD GWAS	Number of SNPs	b	SE	*p*-Value	OR (95% CI)	Cochran’s Q	Heterogeneity Test *p*-Value	MR-Egger Intercept *p*-Value
MR Egger	Sample one	31	−0.17	0.04	0.0005	0.85 (0.78, 0.92)	30.87	0.37	0.37
	Sample two	38	−0.24	0.06	0.0003	0.79 (0.70, 0.89)	42.02	0.23	0.34
	Sample three	43	−0.24	0.08	0.003	0.79 (0.68, 0.91)	33.00	0.80	0.09
Weighted median	Sample one	31	−0.16	0.04	1.4 × 10^−5^	0.85 (0.79, 0.92)	..	..	..
	Sample two	38	−0.22	0.05	1.6 × 10^−5^	0.80 (0.72, 0.89)	..	..	..
	Sample three	43	−0.19	0.07	0.008	0.83 (0.72, 0.95)	..	..	..
Inverse variance weighted	Sample one	31	−0.14	0.03	6.9 × 10^−6^	0.87 (0.82, 0.92)	31.76	0.38	..
	Sample two	38	−0.20	0.04	7.4 × 10^−6^	0.82 (0.75, 0.89)	43.11	0.23	..
	Sample three	43	−0.15	0.06	0.007	0.86 (0.77, 0.96)	36.00	0.72	..
Simple mode	Sample one	31	−0.31	0.11	0.008	0.73 (0.59, 0.91)	..	..	..
	Sample two	38	−0.28	0.14	0.05	0.76 (0.58, 0.99)	..	..	..
	Sample three	43	−0.11	0.16	0.5	0.90 (0.66, 1.23)	..	..	..
Weighted mode	Sample one	31	−0.15	0.04	0.0002	0.86 (0.80, 0.92)	..	..	..
	Sample two	38	−0.21	0.04	2.7 × 10^−5^	0.81 (0.74, 0.88)	..	..	..
	Sample three	43	−0.19	0.07	0.007	0.83 (0.73, 0.94)	..	..	..

## Data Availability

The GWAS summary level data can be obtained via MR-Base. The eQTL summary level data can be obtained via GTEx. UK Biobank data must be requested via www.ukbiobank.ac.uk accessed on 30 May 2022.

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
