# Peer review of "Mendelian Randomization Indicates a Causal Role for Omega-3 Fatty Acids in Inflammatory Bowel Disease"

_ijms, 2022, doi:10.3390/ijms232214380_

Round 1
Reviewer 1 Report
The study by Astone et al. is focused on the causal relationship between ω3 fatty acids and IBD, with implications regarding their effect on other autoimmune diseases such as rheumatoid arthritis.
The manuscript is clearly structured, comprehensible with a minimum of typographical errors. The methods are adequately explained, including supplementary information.
I have only minor comments or recommendations on the work:
I would recommend to put table 1 on a separate page to make the table more clear.
Please move the legendary figure to figure 5 below the figure.
Carefully review the images and align the font size. This is most evident in Figure 4 where, for example, C and F are significantly larger than the other panel numbering.
And last , line 178 move the title of chapter 3.Discussion to the next page
Author Response
Please see our revisions in red.
Reviewer 1:
The study by Astone et al. is focused on the causal relationship between ω3 fatty acids and IBD, with implications regarding their effect on other autoimmune diseases such as rheumatoid arthritis.
The manuscript is clearly structured, comprehensible with a minimum of typographical errors. The methods are adequately explained, including supplementary information.
I have only minor comments or recommendations on the work:
- I would recommend to put table 1 on a separate page to make the table more clear. Table 1 has been moved to a separate page.
- Please move the legendary figure to figure 5 below the figure. We are not sure what the reviewer is requesting, but prefer to retain the locations of the two black boxes.
- Carefully review the images and align the font size. This is most evident in Figure 4 where, for example, C and F are significantly larger than the other panel numbering: Figures 1-3 and 5 were carefully checked for font size differences. The font sizes have been corrected for Figure 4. Thank you for noticing this.
- And last , line 178 move the title of chapter 3.Discussion to the next page: This has been corrected upon moving Table 1 to a new page.
Reviewer 2 Report
Astore, Nagpal and Gibson analyzed the nature of relationship between omega-3 fatty acids and the risk of inflammatory bowel diseases (IBD) by using genetic data from the UK Biobank. This is a much-needed study as new data emerge which point towards the beneficial effects of low-fat diet in ulcerative colitis. The employed methodology is optimal.
Minor comments:
1. IBD is not introduced in the abstract: inflammatory bowel diseases (IBD).
2. Introduction rightly refers to the complexity of interactions between PUFAs and autoimmunity/inflammation. Early in the introduction please briefly refer to the burden of IBD in developed countries and/or globally (incidence or prevalence or economic cost or impact on the quality of life).
3. I am not certain if Results come before Methods in this journal, please verify.
4. “IBD-GC” is not explained at first use. GWAS also. Please make sure that such abbreviations are explained to make the manuscript accessible to all readers.
5. Can resolution of Figure 1 be improved?
6. How were the SNPs selected?
7. The discussion regarding the influence of other variants (than FADS2) on omega-3 fatty acid concentrations is interesting. I imagine this aspect may be considered a limitation of the study but I feel this does not require further clarification if the effect persists despite FADS2 removal (leave-one-out validation).
8. eQTLs from whole blood and the colon were analyzed, even though some of the identified effects were the strongest for Crohn’s disease – this would suggest to make a search for eQTLs in the terminal ileum, which is also available in GTEx. But I think this does not have to be done (this is alluded to in line 128: “FADS2 in blood and terminal ileum”?), but only commented on. In fact, the eQTLs were valid in the blood as well, which should differ from the transverse colon more than the terminal ileum. (Some of the SNPs contained in the grey boxes in Fig. 2 might have significant eQTLs in another tissue)
9. Figure 3. Ratio of linoleic acid to total fatty acids is perplexing to me. Does this concern total LA or gamma-linoleic acid or alpha-linoleic acid? Does it concern the ratio of alpha-linoleic acid to EPA/DHA? I imagine this may be hard to determine – but this might suggest that ALA is not beneficial (?).
10. The analyses of gene x environment interactions are very useful.
11. How were the SNPs selected for MR?
12. The Authors may wish to put gene and transcript names in italics.
Author Response
Please see our revisions in red.
Reviewer 2:
Astore, Nagpal and Gibson analyzed the nature of relationship between omega-3 fatty acids and the risk of inflammatory bowel diseases (IBD) by using genetic data from the UK Biobank. This is a much-needed study as new data emerge which point towards the beneficial effects of low-fat diet in ulcerative colitis. The employed methodology is optimal.
Minor comments:
- IBD is not introduced in the abstract: inflammatory bowel diseases (IBD).
We have included a sentence introducing Inflammatory bowel disease in the abstract. Please note, to meet the 200 word abstract limit, we removed the sentence referring to the differential gut and blood eQTL analysis.
“Inflammatory bowel disease (IBD), is characterized by chronic inflammation of the gastrointestinal system.”
- Introduction rightly refers to the complexity of interactions between PUFAs and autoimmunity/inflammation. Early in the introduction please briefly refer to the burden of IBD in developed countries and/or globally (incidence or prevalence or economic cost or impact on the quality of life).
This is a great point that ties into the question regarding how genetic effects differ across populations. We have added a couple of new sentences at the beginning of the introduction addressing the changing global burden of IBD by reference to a couple of Kaplan papers.
We have added the following reference:
- Kaplan G.G, Windsor J.W., The four epidemiological stages in the global evolution of inflammatory bowel disease. Nat Rev Gastroenterol Hepatol 2021;18:56-66.
All citation numbers have been corrected in the main text and in the supplemental.
“While IBD has been previously thought to only impact the Western world, the occurrence is emerging in developing countries, possibly due to modern environmental exposures [2]. Approximately 1.4 million Americans are impacted by IBD, which may be the result of environmental exposures such as diet and smoking status [3].”
- I am not certain if Results come before Methods in this journal, please verify.
A template provided by IJMS was used to format and organize the text. It is to our understanding that the Results comes before the Methods.
- “IBD-GC” is not explained at first use. GWAS also. Please make sure that such abbreviations are explained to make the manuscript accessible to all readers.
A modification to the first sentence of results section 2.1 was made to define IBD-GC.
To evaluate whether ω3 fatty acids are likely to be causally involved in the pathogenesis of IBD, we identified 31 IBD-associated SNPs in the sample one IBD-Genetics Consortium (GC) GWAS (Table S1), 38 SNPs in the sample two IBD GWAS (Table S2), and subsequently 43 SNPs in the sample three IBD GWAS (Table S3) as genetic instruments.
- Can resolution of Figure 1 be improved?
Unfortunately, if I increase the size of the figure, the font sizes become too small to be legible. However, please not that the PDF resolution allows for zooming in with no pixelation. We will work with the journal to ensure that this is retained.
- How were the SNPs selected?
We have a section in the supplemental materials, “Genetic Instrument Selection criteria” where we describe the details on how the genetic instruments (SNPs) were selected for the MR models.
We modified sentence 3 in section 4.1 Study design to point towards the supplemental materials for details regarding the genetic instrument selection criteria:
“Stringent criteria for genetic instrument selection, as described in the supplemental materials, were implemented in the open source 2SampleMR package in R [30], also supplemented with additional controls.”
Here is the first sentence in the first paragraph of the aforementioned section in the supplemental materials:
“To determine the SNPs to use as genetic instrumental variables for ω3 fatty acids, a genome-wide significance p-value threshold of 5.0 x 10-8 and linkage disequilibrium (LD) clumping was performed, using the 1000 Genomes Project (EUR) as the reference panel, for which an R2 threshold of 0.001 within a 10 Mb interval was applied [7, 35-38].”
- The discussion regarding the influence of other variants (than FADS2) on omega-3 fatty acid concentrations is interesting. I imagine this aspect may be considered a limitation of the study but I feel this does not require further clarification if the effect persists despite FADS2 removal (leave-one-out validation).
Thank you for this thought. The effect does persist upon removal of the FADS2 SNP.
- eQTLs from whole blood and the colon were analyzed, even though some of the identified effects were the strongest for Crohn’s disease – this would suggest to make a search for eQTLs in the terminal ileum, which is also available in GTEx. But I think this does not have to be done (this is alluded to in line 128: “FADS2 in blood and terminal ileum”?), but only commented on. In fact, the eQTLs were valid in the blood as well, which should differ from the transverse colon more than the terminal ileum. (Some of the SNPs contained in the grey boxes in Fig. 2 might have significant eQTLs in another tissue)
Supplemental table 4 (Table S4) demonstrates other tissue eQTLs, including the terminal ileium, for the FADS1/2 SNP, rs174564. In the text, we have pointed to Table S4 in the discussion, “Notably, the largest eQTL effect sizes were consistently observed for FADS2, which encodes a fatty acid desaturase, and showed the largest GWAS association with IBD. Since FADS1 has also been implicated in IBD risk [16], we further contrasted the eQTL effects of the lead GWAS variant at the FADS1/2 locus, rs174564, on both genes, finding much stronger association with FADS2 in blood and terminal ileum, and similar effect sizes but in opposite directions in the colon (Table S4).”
|
Table S4: rs174564 FADS1/FADS2 eQTL summary results for the whole blood, transverse colon, sigmoid colon, liver and small intestine - terminal ilieum. |
||||
|
gene |
SNP |
P-Value |
NES |
Tissue |
|
FADS1 |
rs174564 |
0.0000013 |
0.19 |
Whole Blood |
|
FADS1 |
rs174564 |
0.0064 |
-0.07 |
Colon - Transverse |
|
FADS1 |
rs174564 |
2.20E-10 |
-0.29 |
Colon - Sigmoid |
|
FADS1 |
rs174564 |
4.30E-07 |
-0.41 |
Liver |
|
FADS1 |
rs174564 |
0.19 |
0.061 |
Small Intestine - Terminal Ileum |
|
FADS2 |
rs174564 |
5.40E-54 |
0.72 |
Whole Blood |
|
FADS2 |
rs174564 |
9.70E-16 |
0.29 |
Colon - Transverse |
|
FADS2 |
rs174564 |
5.10E-08 |
0.28 |
Colon - Sigmoid |
|
FADS2 |
rs174564 |
0.056 |
-0.14 |
Liver |
|
FADS2 |
rs174564 |
3.60E-11 |
0.58 |
Small Intestine - Terminal Ileum |
- Figure 3. Ratio of linoleic acid to total fatty acids is perplexing to me. Does this concern total LA or gamma-linoleic acid or alpha-linoleic acid? Does it concern the ratio of alpha-linoleic acid to EPA/DHA? I imagine this may be hard to determine – but this might suggest that ALA is not beneficial (?).
Unfortunately, there is no field in the UKBB for contrasting gamma and alpha-linoleic acid, only linoleic acid, so we cannot comment.
- The analyses of gene x environment interactions are very useful.
Thank you.
- How were the SNPs selected for MR?
Please see response for point 6.
- The Authors may wish to put gene and transcript names in italics.
All the gene names have been italicized.
Reviewer 3 Report
1. One part I wish the authors could improve on was the detailed description of the study and validation populations. For example, how many samples are there, and are there any differences in their compositions? Importantly, it is not shown in the manuscript how you reached the number of candidate instrument variants. It would be helpful to include a description of the quality control steps regarding the number of SNPs removed/kept.
2. In Figure 5, in the genetic instrument selection criteria, it is unclear whether you used the p-value threshold for genotype-outcome association or genotype-exposure association.
3. This comment is related to comment #1. As you mentioned that omega-3 fatty acids are largely obtained from the diet, have you considered the possibility of reverse causality? Is it possible that the IBD patients changed their diet after the diagnosis? I am not an expert on this and would greatly benefit from a more detailed description of the study populations and study design of the underlying GWAS study. Additionally, have you tried to investigate the possibility of reverse causation using mendelian randomization?
4. In lines 79 and 84, it would be clearer to briefly mention what types of MR methods you used instead of “other MR methods”.
5. Standardized names for the data sets used in the study should be provided.
